# A Novel Shape-Prior-Guided Automatic Calibration Method for Free-Hand Three-Dimensional Ultrasonography

**DOI:** 10.3390/s25165104

**Published:** 2025-08-17

**Authors:** Xing-Yang Liu, Jia-Xu Zhao, Hui Tang, Guang-Quan Zhou

**Affiliations:** 1The School of Biological Science and Medical Engineering, Southeast University, Nanjing 211189, China; 2School of Computer Science and Engineering, Southeast University, Nanjing 211189, China; corinna@seu.edu.cn

**Keywords:** ultrasound probe calibration, N-wire phantom, automatic feature extraction, RANSAC-based outlier rejection

## Abstract

Ultrasound probe calibration is crucial for precise spatial mapping in ultrasound-guided surgical navigation and free-hand 3D ultrasound imaging as it establishes the rigid-body transformation between the ultrasound image plane and an external tracking sensor. However, the existing methods often rely on manual feature point selection and exhibit limited robustness to outliers, resulting in reduced accuracy, reproducibility, and efficiency. To address these limitations, we propose a fully automated calibration framework that leverages the geometric priors of an N-wire phantom to achieve reliable recognition. The method incorporates a robust feature point extraction algorithm and integrates a hybrid outlier rejection strategy based on the Random Sample Consensus (RANSAC) algorithm. The experimental evaluations demonstrate sub-millimeter accuracy (<0.6 mm) across varying imaging depths, with the calibration process completed in under two minutes and exhibiting high repeatability. These results suggest that the proposed framework provides a robust, accurate, and time-efficient solution for ultrasound probe calibration, with strong potential for clinical integration.

## 1. Introduction

Ultrasound imaging, compared to modalities such as MRI and CT, offers distinct advantages including real-time visualization, absence of ionizing radiation, and portability [1], making it particularly suitable for image-guided interventional procedures. Consequently, ultrasound-guided surgical navigation systems have demonstrated strong potential in applications such as ablation therapy, evolving into essential tools for intraoperative guidance [2]. A critical requirement in such systems is accurate spatial localization of ultrasound images, which serves as the foundation for effective visual navigation. However, as ultrasound inherently provides only two-dimensional information, the spatial pose (position and orientation) of each image must be determined through a spatial tracking system that is rigidly attached to the probe. Due to a fixed structural offset between the electromagnetic (EM) sensor and the imaging plane, a rigid transformation matrix must be calibrated to map the sensor’s coordinate system to that of the ultrasound image. This transformation is typically computed via a dedicated ultrasound probe calibration process, enabling the conversion of 2D image pixels into corresponding 3D spatial coordinates [3].

Most existing ultrasound probe calibration methods rely on physical phantoms with known geometric structures to establish spatial correspondences between the ultrasound image plane and the phantom [4]. Among the various phantoms, the N-wire design is widely adopted due to its high density of intersection points, ease of segmentation, and capability to achieve high calibration accuracy with limited data [5,6,7]. Many prior works have proposed different strategies using N-wire phantoms. For instance, Chen et al. [8] developed a fully automatic segmentation algorithm for detecting intersection points of double N-wire structures, along with a real-time accuracy control mechanism. Wen et al. [9] introduced a method that combines N-wire-based calibration with stylus registration, utilizing a Gaussian model for feature extraction and iterative closest point (ICP) optimization for matrix estimation.

Based on the above phantoms, a standard calibration pipeline involves three key stages: acquisition of calibrated images, extraction of feature points from these images, and computation of the transformation matrix. Although many studies have achieved calibration based on this three-stage process, these studies still have certain deficiencies in the two key links: image feature point extraction and estimation of coordinate transformation parameters. In terms of image feature extraction, most existing methods still rely on manual annotation of feature points, resulting in low calibration efficiency and relatively high errors [6,10,11]. Although some studies have achieved automatic feature point extraction [8,12,13,14], these approaches are often sensitive to image quality [15,16] or fail to utilize the prior geometric structure of the calibration phantom effectively [8,12]. As a result, they are prone to false detections or missed detections in the presence of speckle noise, feature ambiguity, and other artifacts common in ultrasound images. Furthermore, in terms of solving the transformation matrix, most methods adopt least-squares [17,18] or closed-form solutions [13]. However, these approaches generally lack robust outlier rejection mechanisms, making them vulnerable to practical sources of error such as inaccurate feature localization or tracking system noise. Such disturbances can cause some reference points to deviate from their true spatial positions, introducing outliers that ultimately degrade the accuracy of the estimated calibration matrix.

Recent years have seen continued efforts to improve the automation, accuracy, and robustness of ultrasound calibration. Wu et al. [12] proposed a fully automatic method using a multilayer N-wire phantom, achieving sub-millimeter accuracy but without outlier filtering. Shen et al. [19] proposed a method based on a general wire phantom and closed-form solution, which offers flexibility in phantom design but relies on manual feature identification, limiting its efficiency. In addition to the traditional ultrasound probe calibration, robot-assisted calibration methods have also emerged in recent years. They employ stylus tracking or geometric phantom alignment (e.g., sphere phantoms) [20], Z-fiducial phantoms [21], and M-fiducial phantoms [22]. While these approaches aim to automate the calibration workflow, many rely on simple geometric fitting or manual thresholding for feature extraction, which are prone to false detections when imaging artifacts are present [13]. For example, some methods use circular or linear fitting without leveraging full geometric priors, leading to inaccuracies at increased depths or under noisy conditions [22,23]. Additionally, stylus-based methods may require manual intervention or are limited by constrained scan trajectories [20,23]. Furthermore, many existing approaches lack robust outlier rejection mechanisms during feature point selection or spatial transformation estimation, which can adversely affect their accuracy and repeatability.

To address these limitations, this paper proposes a fully automatic ultrasound probe calibration method that integrates feature point auto-detection with robust outlier rejection. The method leverages prior geometric constraints of the N-wire phantom structure to automatically extract feature points and introduces a RANSAC-based outlier removal mechanism [24] during the spatial transformation estimation stage to enhance accuracy and robustness. The experimental results demonstrate that the proposed method can achieve sub-millimeter accuracy and high repeatability across multiple imaging depths and probe types, offering a reliable and time-efficient solution suitable for clinical deployment.

## 2. Materials and Methods

### 2.1. Probe Calibration Overview

To estimate the calibration matrix between the ultrasound image and the tracking sensor, this study introduces four coordinate systems: the ultrasound image coordinate system (Ultrasound, U), the sensor coordinate system (Receiver, R), the electromagnetic tracking system coordinate system (World, W), and the N-wire phantom coordinate system (Phantom, P). The transformation relationships among these coordinate systems are illustrated in Figure 1.

In Figure 1, let $T_{A}^{B}$ represent the homogeneous transformation matrix that maps a point from coordinate system A to coordinate system B. The transformation matrix from the ultrasound image coordinate system U to the sensor coordinate system R, denoted as $T_{U}^{R}$, is the calibration matrix to be estimated. To compute this matrix, each pixel in the ultrasound image (defined in coordinate system U) must be mapped to its corresponding spatial location in the phantom coordinate system P. The objective is to minimize the Euclidean distance between the transformed point and its ground-truth position in P, thereby establishing an accurate spatial correspondence between the image and the phantom. This formulation gives rise to the following nonlinear optimization problem:(1)$$\underset{{T_{U}^{R},T}_{W}^{P},s_{x},s_{y}}{\min}\sum_{i=1}^{N}{\left\|X_{i}^{p}-T_{W}^{P}{T_{R}^{W}T}_{U}^{R}X_{i}^{U}\right\|}_{2}$$
Let $X_{i}^{p}={\left(x_{i}^{p},y_{i}^{p},z_{i}^{p},1\right)}^{T}$ denote the 3D homogeneous coordinate of the *i*-th feature point in the phantom coordinate system P, and let $X_{i}^{U}={\left({s_{x}u}_{i},s_{y}v_{i},0,1\right)}^{T}$ denote the corresponding homogeneous coordinate in the ultrasound image coordinate system U, where ${(u}_{i}$, $v_{i})$ is the pixel coordinate of the *i*-th feature point, and $s_{x}$ and $s_{y}$ represent the pixel resolution in the horizontal and vertical directions, respectively (in mm/pixel). The transformation matrix $T_{R}^{W}$ denotes the pose of the tracking sensor in the world coordinate system and is treated as a known quantity obtained from the electromagnetic tracking system. In contrast, the matrix $T_{W}^{P}$, which describes the spatial relationship between the phantom and the tracking system, is treated as an unknown variable. Accordingly, the probe calibration problem involves the following unknowns: $T_{R}^{W}$, $T_{W}^{P}$, $s_{x}$, and $s_{y}$. Both $T_{R}^{W}$ and $T_{W}^{P}$ are 6-DoF rigid body transformations, each parameterized by three Euler angles (rotation) and three translation components. Therefore, the calibration problem is formulated as a 14-parameter nonlinear optimization, which is subsequently solved using the Levenberg–Marquardt (L-M) algorithm [25].

To compute the 3D coordinates of each feature point in the phantom coordinate system, the geometric configuration of the N-wire phantom is exploited. As illustrated in Figure 2, the endpoint a of the N-wire structure is defined as the origin of the phantom coordinate system. The feature point B lies in the same xy plane as a, and thus its z-coordinate is known and fixed. Given that lines AC and BD are parallel and B lies along the middle oblique wire, the coordinates of B can be determined using the principle of similar triangles, as formulated below:(2)$$\alpha \;=\frac{{\left\|B^{U}-A^{U}\right\|}_{2}}{{\left\|C^{U}-A^{U}\right\|}_{2}}=\frac{\left|{x_{B}}^{P}\right|}{{\left\|a^{P}-b^{P}\right\|}_{2}}=\frac{\left|{y_{B}}^{P}\right|}{{\left\|a^{P}-c^{P}\right\|}_{2}}$$
where α is the relative distance ratio computed in the image plane; ${\left\|\cdot \right\|}_{2}$ denotes the Euclidean distance between two points; and $\left|\cdot \right|$ represents the absolute value. The points $a^{P}$, $b^{P}$, $c^{P}$, and $d^{P}$ denote the known vertex positions in the phantom coordinate system, while $A^{U}$, $B^{U}$, and $C^{U}$ represent the corresponding reference points extracted from the ultrasound image coordinate system.

### 2.2. Feature Point Extraction Method

(1)Feature Point Shape Prior Based on Multi-Scale Hessian Response

To enhance the detection of N-wire intersection points in ultrasound images, this study incorporates a combination of morphological and differential operators guided by feature shape priors.

Due to the influence of the point spread function (PSF), intersections between the ultrasound beam and the N-wire structure manifest as bright, localized blob-like regions in the image. While this characteristic facilitates detection, high-intensity reflections from large-scale structures—particularly at the phantom’s boundaries—can obscure the target features. To address this, a top-hat transformation is applied, which subtracts the morphological opening result from the original image. This operation suppresses background structures larger than the target features while preserving localized bright spots. As shown in Figure 3, the red rectangles in panel (a) indicate elongated echo artifacts caused by structural reflections, while panel (b) demonstrates the effectiveness of the top-hat transformation using directionally oriented rectangular structuring elements at angles of 0°, 45°, 90°, and 135°. Compared to circular structuring elements, the directional approach more effectively attenuates anisotropic noise.

After top-hat processing, candidate features generally appear as circular or elliptical blobs. To further enhance localization, a Hessian matrix-based blob detection mechanism is introduced. The Hessian matrix characterizes the second-order local intensity variations in an image and serves as a differential descriptor of the geometric structure. Its determinant, which quantifies the response magnitude, is particularly sensitive to blob-like patterns and corner features. By emphasizing regions with high structural symmetry and sharp intensity transitions, it is well-suited for detecting bright, centrally symmetric features commonly found in ultrasound images [26].

However, the effectiveness of Hessian-based feature detection critically depends on the alignment between the filter scale and the apparent size of the target structure in the image. When the filter scale matches the spatial scale of the target feature, the response is maximized; otherwise, the response weakens, reducing the detection accuracy. In clinical ultrasound imaging, the system’s imaging depth is typically adjusted based on the region of interest. This adjustment alters the projected scale of anatomical structures in the image, causing features of identical physical size to appear at varying spatial scales. As a result, fixed-scale filters are insufficient for consistent detection across different imaging depths.

To address this limitation, a multi-scale Hessian response mechanism [27] is adopted in this study. A scale-space representation is constructed, wherein the Hessian response is computed across a range of scales and subsequently normalized and fused. This multi-scale strategy enables the algorithm to maintain stable and discriminative responses for target features under varying depth conditions, ensuring robust scale-invariant detection. At a given image location (x,y), the Hessian matrix at scale σ is defined as(3)$$H\left(x,y,\sigma \right)=\left[\begin{array}{cc} \begin{array}{c} I_{xx}\left(x,y,\sigma \right)\\ I_{xy}\left(x,y,\sigma \right) \end{array} & \begin{array}{c} I_{xy}\left(x,y,\sigma \right)\\ I_{yy}\left(x,y,\sigma \right) \end{array} \end{array}\right]$$
where $I_{xx}$, $I_{xy}$, and $I_{yy}$ denote the second-order partial derivatives of the image intensity at scale σ, which are computed using scale-normalized Gaussian derivative filters to ensure consistency across multiple scales. The scale parameter σ is selected to reflect the expected variation in the apparent size of target features under different ultrasound imaging depths (typically 5–12 cm). It effectively captures the target feature points while avoiding spatial blurring and localization errors caused by scales that are excessively large.

To ensure the comparability of the Hessian responses across scales, a normalization factor $\sigma^{4}$ is introduced, yielding the scale-normalized determinant:(4)$${det}_{\sigma }\left(H\right)=\left(I_{xx}\cdot I_{yy}-{I_{xx}}^{2}\right)\times \sigma^{4}$$

This determinant value reflects the local structural saliency at a given scale: a higher response indicates the presence of centrally symmetric, blob-like features that are geometrically distinct from the surrounding background. To aggregate information across scales, the maximum response is selected for each image location (x,y):(5)$$\Lambda \left(x,y\right)=\underset{\sigma \in S}{\max}\;{det}_{\sigma }\left(H\left(x,y\right)\right)$$
A global threshold $\delta =0.04\cdot max\left(\Lambda \right)$ is applied to suppress weak responses, and the response map is binarized as follows:(6)$$\Lambda \left(x,y\right)=\left\{\begin{array}{ll} 0, & \Lambda \left(x,y\right)<\delta \\ \Lambda \left(x,y\right), & otherwise \end{array}\right.$$

Subsequently, 8-connected non-maximum suppression is applied to Λ to identify local maxima, forming the candidate feature point set P. This method combines scale-normalized Gaussian derivative kernels and multi-scale response fusion, enabling robust and scale-invariant detection of feature points. The extracted feature candidates, marked as green dots in Figure 4, are then passed to the geometric constraint filtering stage for further refinement.

(2)Feature Point Filtering Based on N-Wire Structural Prior

As illustrated in Figure 4, the candidate point set P obtained via shape-based filtering still contains non-target bright points that are similar in size and intensity to valid features. To further refine this set, we incorporate geometric structure priors derived from the physical configuration of the N-wire phantom. This phantom comprises multiple layers of regularly arranged nylon wires that form a patterned, mesh-like structure in ultrasound images. Specifically, three types of geometric regularities are observed: (1) intra-layer collinearity—three feature points within the same layer lie approximately on a straight line; (2) inter-layer parallelism—the lines formed by adjacent layers of collinear points exhibit approximate parallelism; and (3) distance consistency—feature point spacings are relatively uniform both within and between layers.

Based on these priors, the filtering process proceeds in three stages. First, intra-layer collinearity is enforced by iterating through point triplets in P. For each triplet, the slopes $k_{1}$ and $k_{2}$ between the middle point and its neighbors are calculated. If both $|k_{1}|\;<\;\varepsilon$ and $|k_{2}|\;<\;\varepsilon$ are satisfied, the triplet is considered collinear and retained. Next, to assess inter-layer parallelism, the slopes k of all retained collinear triplets are sorted to form a sequence S. A sliding window of size W, equal to the number of wire layers, is applied, and the segment with the smallest slope variation is selected as(7)$$i=argmin\left(S\left[i+W-1\right]-S\left[i\right]\right)$$
where i denotes the window index in S. The average slope within this window is then defined as the main slope $k_{main}$ of the phantom. All other collinear triplets are then compared to $k_{main}$ and only retained if their slope difference satisfies $|k\;-\;k_{main}|\;<\varepsilon_{1}$.

Finally, distance consistency constraints are applied to further refine the candidate feature points. While the application of collinearity and parallelism priors improves the structural regularity of the selected point set, residual false positives may still persist. As illustrated in Figure 4, the points highlighted with red circles satisfy both the collinearity and parallelism criteria but deviate from the expected spatial distribution and are thus identified as false detections.

Figure 5 shows a schematic of the N-wire phantom, where yellow triangles represent parallel nylon wires and red triangles denote oblique elements. The inter-wire and inter-layer distances are physically fixed in the phantom structure, and this spatial regularity should be preserved in the corresponding image-space point set. To enforce this constraint, the intra-layer and inter-layer pairwise distances between adjacent feature points are evaluated, and each candidate triplet is required to satisfy the following consistency condition:(8)$$\left\{\begin{array}{c} \left|x_{j+1}^{i}-x_{j}^{i}\right|\;-\;\left|x_{j+2}^{i}-x_{j+1}^{i}\right|\;\leq \;diff\\ \left|x_{j}^{i+1}-x_{j}^{i}\right|\;-\;\left|x_{j}^{i+2}-x_{j}^{i+1}\right|\;\leq \;diff \end{array}\right.$$
where $x_{j}^{i}$ denotes the *j*-th feature point in the *i*-th layer, and diff is the predefined tolerance threshold. The first condition enforces intra-layer distance consistency, while the second applies to inter-layer spacing. Only points satisfying both conditions are retained as true N-wire feature points; all others are considered false positives and excluded.

(3)Subpixel Center Localization of Feature Points

After applying shape- and structure-based priors, the approximate positions of feature points in the ultrasound image are obtained. However, due to the presence of speckle noise and variability in local intensity distributions, these initial positions may deviate from the actual centers of the target regions. To improve localization precision, this study performs subpixel refinement based on local intensity statistics, as illustrated in Figure 6. Given an initial feature point at $\left(x_{0},\;y_{0}\right)$, a square search window with radius r is defined, and candidate pixels are selected within the region $x\;\in \;[x_{0}-r,x_{0}\;+\;r]$, $y\in [y_{0}\;-\;r,y_{0}\;+\;r]$. For each pixel within this region, the average intensity of its 8-connected neighborhood is computed. The pixel whose neighborhood exhibits the highest average intensity is selected as the refined center of the corresponding high-intensity region, and is subsequently used as the final feature point location. 

### 2.3. Robust Optimization via RANSAC-Assisted Levenberg–Marquardt Algorithm

To improve robustness against outliers caused by electromagnetic tracking noise, ultrasound image degradation, or phantom fabrication inaccuracies, a hybrid optimization strategy combining RANSAC and L-M algorithm is adopted. RANSAC identifies inliers by iteratively sampling minimal subsets of point correspondences and estimating temporary models. In each iteration, a random subset S (typically >5 correspondences) is selected to compute a preliminary transformation. The L-M algorithm is then applied to solve the nonlinear calibration model based on this subset.

Residual errors are computed for all points, and inliers are identified as those satisfying a predefined threshold. Specifically, the residual for the *i*-th correspondence is defined as(9)$$r_{i}={\left\|X_{i}^{P}-{\hat{T}}_{W}^{P}T_{R}^{W}{\hat{T}}_{U}^{R}{\hat{X}}_{i}^{U}\right\|}_{2}$$
where $r_{i}$ denotes the Euclidean residual between the transformed ultrasound point and its corresponding ground-truth position in the phantom space. For detailed definitions of the transformation matrices and point notations, please refer to Equation (1).

Points with an ${r_{i}<\varepsilon }_{2}$ are treated as inliers. The model yielding the largest inlier set is retained. Once a satisfactory inlier ratio is achieved, RANSAC terminates, and the final inlier set is passed to the L-M algorithm for refinement of the calibration parameters.

This combined strategy leverages the robustness of RANSAC with the accuracy of L-M optimization. To improve efficiency, termination criteria are set based on the number of iterations and convergence of the inlier set. For implementation details and theoretical background, readers are referred to [24,25,28,29].

### 2.4. Experiment Design

To ensure reproducibility and transparency, detailed specifications of the experimental setup are provided as follows. Two ultrasound imaging systems were used in this study: the Philips EPIQ 5 (Philips Healthcare, Amsterdam, The Netherlands) and the NeuBook 3 (Neusoft Medical Systems Co., Ltd., Shenyang, China). Two types of commonly used ultrasound probes were employed: a convex array probe (C5-2) and a linear array probe (L12-5). The ultrasound video was captured using an HD60 S+ HDMI/USB 3.0 acquisition card with a frame rate of 60 frames per second. An electromagnetic tracking system (NDI Aurora, Northern Digital Inc., Waterloo, ON, Canada) which was available from our laboratory inventory, was utilized to obtain 6-DOF spatial pose data at a sampling frequency of 40 Hz. The system has a spatial tracking accuracy better than 1 mm. A miniature EM sensor was rigidly affixed to the ultrasound probe using a custom 3D-printed holder. The calibration phantom used in the study was an open-source N-wire design obtained from the Plus Toolkit project. It was fabricated using a 3D printer, with a vertical spacing of 10 mm between layers, 25 mm spacing between parallel wires, and nylon wires with a diameter of 0.3 mm. All image and pose data were processed and stored on a workstation running Windows 11 (64-bit), equipped with an Intel Core i7-12700H CPU (Intel Corporation, Santa Clara, CA, USA), an NVIDIA RTX 3060 GPU (NVIDIA Corporation, Santa Clara, CA, USA), and 16 GB of RAM. The algorithmic development environment included C++ (Microsoft Visual Studio 2022), OpenCV 4.8.0, Eigen 3.8, MATLAB R2022b, and NDI SDK 3.1.

To simulate realistic clinical conditions, all experiments were conducted in a transparent acrylic water tank containing three sets of N-wire phantoms fixed to the base using nylon wires. The water temperature was maintained between 36 and 37 °C to approximate the thermal environment of human tissue. Data acquisition was independently performed by three experienced ultrasound operators, using two types of ultrasound probes—a convex array and linear array—and four commonly used imaging depths (5 cm, 8 cm, 10 cm, and 12 cm). For each depth, a total of 10 datasets were collected: 5 using the convex array probe and 5 using the linear array probe. Each dataset consisted of 30 ultrasound frames along with corresponding electromagnetic pose data, which were used for feature point extraction, calibration accuracy evaluation, and repeatability validation. During acquisition, the probe was manipulated with multi-axis rotations (including pitch and yaw) and moderate translations above the phantom to introduce variability in spatial pose and imaging characteristics, thereby increasing the diversity of the data and avoiding convergence to local optima during optimization.

The algorithmic parameters were empirically set as follows. The external search radius r for subpixel localization was set to 30 pixels. Both the collinearity threshold ε and the inter-layer parallelism threshold $\varepsilon_{1}$ were set to 0.05. The distance consistency constraint threshold diff was set to 10 pixels. For RANSAC-based outlier rejection, the number of iterations was set to 3000, the minimum subset size was set to 10, and the residual error threshold $\varepsilon_{2}$ was set to 0.5.

## 3. Results

### 3.1. Feature Point Detection Accuracy

To evaluate the robustness, generalizability, and cross-device adaptability of the proposed feature point detection algorithm, controlled experiments were performed using two different ultrasound systems: NeuBook 3 and Philips EPIQ 5. For each system, ultrasound images were acquired at three representative imaging depths (5 cm, 8 cm, and 12 cm), and the evaluation focused on three key performance metrics: recognition rate, false detection rate, and processing time per image set. The recognition rate and false detection rate are defined, respectively, as follows:(10)$$Recognition\;Rate\;=\;\frac{N_{detected}}{N_{total}}$$(11)$$False\;Detection\;Rate=\frac{N_{false}}{N_{false}+N_{detected}}$$
where $N_{detected}$ is the number of correctly identified feature points, $N_{false}$ is the number of incorrectly identified non-feature points, and $N_{total}$ is the total number of ground-truth feature points.

Table 1 presents the detection performance at varying ultrasound imaging depths for both convex array and linear array probes across the two different ultrasound systems. The recognition rate remained consistently high across all conditions, ranging from 93.333% to 99.000%, indicating the method’s strong adaptability to diverse imaging conditions and device types. Notably, the false detection rate remained at 0% in all cases, which is not explicitly reported in the table for brevity. In terms of computational efficiency, the average processing time per image set ranged from 9.45 s to 15.72 s, with minor variations depending on the probe type and system used. All measured times support the method’s suitability for real-time or near-real-time clinical workflows.

Furthermore, the consistent performance across both ultrasound platforms and probe types, along with data collected independently by three experienced operators, confirmed the robustness and generalizability of the proposed algorithm with respect to both hardware variability and operator differences. Representative detection outcomes are shown in Figure 7a.

To evaluate the contribution of each algorithmic component, ablation studies were conducted using a convex array probe at 5 cm and 12 cm depths. The results are summarized in Table 2.

First, the impact of removing the top-hat transformation is visualized in Figure 7b. Without this preprocessing step, the number of candidate points increased substantially due to residual background structures, leading to a significant rise in computation time—exceeding 60 s per frame—which renders the method impractical for real-time use. Given its critical role in background suppression, this module was not further evaluated in the ablation table.

Next, the multi-scale Hessian matrix was replaced with a conventional connected component labeling approach. As shown in Table 2, the recognition rate notably declined at a depth of 12 cm, demonstrating the necessity of scale-invariant response mechanisms. The multi-scale Hessian enables adaptive feature point localization across variable imaging depths, addressing the limitations of fixed-threshold or single-scale methods.

Finally, the geometric constraint module based on the N-wire phantom structure was removed. This led to a marked performance drop, with recognition rate decreasing from 98.215% to 87.333% and false detection rate increasing to 12.21%. As shown in Figure 7c, several incorrect feature points satisfied the collinearity and parallelism criteria but failed to meet the distance consistency constraint, underscoring the necessity of combining all geometric priors.

These results collectively demonstrate that each algorithmic component—top-hat transformation, multi-scale Hessian filtering, and geometric priors—contributes significantly to the overall detection accuracy. The ablation study confirmed that only their integrated use yields optimal performance. As summarized in Table 2, omission of any single module resulted in reduced recognition rates and increased susceptibility to false positives, especially under challenging imaging depths.

### 3.2. Automatic Calibration Accuracy

To quantitatively evaluate the accuracy of the proposed automatic ultrasound probe calibration method and assess the effectiveness of the RANSAC-based outlier rejection strategy, the Euclidean distance between the transformed feature points and their corresponding ground-truth positions in the phantom coordinate system was employed as the evaluation metric. After solving the matrices $T_{U}^{R}$ and $T_{W}^{P}$, each detected feature point ${\;X}_{i}^{U}$ in the ultrasound image coordinate system was mapped to the phantom coordinate system, and the residual was obtained by subtracting it from the ground-truth position:(12)$$e_{i}={\left\|X_{i}^{P}-T_{W}^{P}T_{R}^{W}T_{U}^{R}X_{i}^{U}\right\|}_{2}$$
and the overall calibration accuracy was then summarized using the following statistical indicators:(13)$$mean\;=\frac{1}{N}\sum_{i=1}^{N}e_{i},max=max\left(e_{i}\right)$$

As summarized in Table 3, the proposed method was tested under four imaging depths (5 cm, 8 cm, 10 cm, and 12 cm), using both the conventional L-M-only optimization and the hybrid L-M + RANSAC strategies. The results showed that L-M + RANSAC consistently achieved sub-millimeter accuracy, with the mean errors ranging from 0.497 mm to 0.571 mm and the maximum errors remaining within 0.76 mm across all depths. Notably, the smallest mean errors were observed at 5 cm and 8 cm (0.497 mm and 0.499 mm, respectively), which were attributed to higher image contrast and lower speckle noise at shallow imaging depths. At 10 cm, ultrasound attenuation and noise led to a slight increase in error (mean: 0.571 mm; max: 0.758 mm), while at 12 cm, the error decreased again (mean: 0.524 mm; max: 0.542 mm), suggesting improved stability in deeper imaging. In comparison, the L-M-only configuration resulted in higher calibration errors at all depths. The L-M + RANSAC strategy reduced the mean error by 0.20–0.30 mm and the maximum error by 0.25–0.40 mm, confirming the efficacy of the outlier rejection mechanism in improving calibration precision.

Figure 8 illustrates the distribution of calibration errors across the four imaging depths using box plots. At 5 cm and 8 cm, the boxes and whiskers are compact, indicating low variance and consistent calibration results. The median lines lie below 0.60 mm, consistent with the reported mean values. At a 10 cm depth, the interquartile range and whisker length increased significantly, and several outliers (indicated by red crosses) emerged, reflecting higher sensitivity to noise at greater depths. Nevertheless, the RANSAC mechanism successfully bound the median error within an acceptable range. At 12 cm, the error distribution stabilized, with a reduced spread and no significant outliers.

These observations demonstrate that the proposed calibration framework exhibits strong robustness across varying depths and imaging conditions. The integration of RANSAC not only enhances the resistance to noise and outliers but also ensures that the calibration error consistently remains within sub-millimeter accuracy, validating the method’s applicability in real-world clinical scenarios.

### 3.3. Automatic Calibration Repeatability

To assess the repeatability of the proposed automatic calibration method across multiple independent calibration trials, a quantitative verification experiment was designed based on the spatial variation of transformed feature points. A repeatability metric, referred to as Calibration Reproducibility (CR), was introduced to quantify the positional stability of corner points under different calibration results.

For a given corner point, the transformation matrix $T_{U}^{R}$ estimated from each individual calibration session was applied to transform the point from the image coordinate system to the sensor coordinate system, yielding a set of transformed 3D coordinates. The centroid of this set was computed and used as a reference. The CR value for the point was then defined as the average Euclidean distance between each transformed instance and the centroid, reflecting its spatial dispersion. This procedure was repeated for the other three corner points, and the final metric, termed Calibration Reproducibility Error (CRE), was calculated as the mean CR value across all four corner points. The formulation of CRE is given by(14)$$CRE=\frac{1}{4N}\sum_{k=1}^{4}\sum_{i=1}^{N}{\left\|x_{i,k}-\bar{x_{k}}\right\|}_{2}$$
where $x_{i,k}$ denotes the 3D coordinate of the k-th corner point in the *i*-th calibration group, $\bar{x_{k}}$ represents the centroid of that corner point across all trials, and N is the total number of calibration groups. A smaller CRE value indicates a higher calibration consistency and repeatability.

Table 4 presents the CRE statistics at four imaging depths: 5 cm, 8 cm, 10 cm, and 12 cm. Across all depths, the mean CRE remained below 2.2 mm, and the maximum did not exceed 2.3 mm, indicating that the proposed method exhibited high consistency over repeated calibrations. In particular, depths of 5 cm and 12 cm yielded the most stable results, with mean CRE values of 1.121 mm and 1.092 mm, respectively. At 8 cm, the mean CRE was slightly elevated, with the maximum reaching 2.105 mm, potentially due to increased sensitivity to image noise or signal variation at this depth. For 10 cm, both the mean and maximum CRE values remained moderate, indicating a slight increase in fluctuation but still within acceptable bounds.

To further illustrate the distribution characteristics of calibration errors at different depths, Figure 9 presents the CRE boxplots. At 5 cm, despite a few outliers, the box length is the shortest, indicating minimal variation and excellent calibration stability. The boxplot for a depth of 8 cm displayed the widest spread, with the upper whisker approaching 2.1 mm, consistent with the higher mean CRE observed in Table 4. This suggests greater susceptibility to error at this depth. At 10 cm, the spread narrowed slightly, reflecting moderate error fluctuation. Overall, the boxplots across all depths demonstrated tight clustering of calibration errors with minimal dispersion and few significant outliers.

These results confirmed that the proposed method achieved robust and repeatable calibration performance under varying imaging depths, demonstrating its potential for reliable deployment in clinical scenarios.

### 3.4. Comparison with Representative Methods

To validate the advantages of the proposed ultrasound calibration method over other representative semi-automated and automated approaches, we tested two mainstream N-wire phantom-based calibration methods for comparison, namely AGC-FD (Automatic Gray-weighted Centroid Feature Detection) and TACD (Threshold-Assisted Centroid Detection).

The AGC-FD method, proposed by Rong et al. [30], combines adaptive thresholding with gray-level weighted centroid extraction for localized feature detection. This method is fully automatic and emphasizes adaptive image segmentation; however, it is sensitive to noise and lacks geometric structural constraints, resulting in elevated false detection rates. The TACD method, derived from the work of Lei et al. [13], integrates threshold-based segmentation with geometric center localization via minimum enclosing circle fitting. As the threshold selection in TACD typically requires manual or experience-based adjustment, it is categorized as a semi-automated method. Similar to AGC-FD, it does not exploit the geometric priors of the N-wire phantom for false-positive suppression.

During the experiments, all three methods were evaluated using three independent datasets, each consisting of ultrasound images acquired at different imaging depths (5 cm, 8 cm, and 12 cm). The number of images in each dataset was 28, 28, and 17, respectively. The key evaluation metrics included recognition rate, false detection rate, and processing time per image group, as defined in Section 3.1. As summarized in Table 5, our method achieved the highest average recognition rate (98.16%) with zero false detections across all datasets. In contrast, AGC-FD and TACD yielded lower average recognition rates (91.87% and 93.41%, respectively) and significantly higher false detection rates (7.94% and 6.08%, respectively), demonstrating a considerable performance gap compared to our approach.

Figure 10 presents a comparative visualization of the detection results. Analysis of the detection pipelines revealed that both AGC-FD and TACD apply simple Gaussian filtering followed by gray-level thresholding for segmentation. However, these methods failed to address the elongated echo artifacts commonly observed in ultrasound images, leading to false detections, as shown in Figure 10c. Furthermore, although valid feature points typically exhibit elliptical or circular geometry, neither method utilizes dedicated geometric operators for blob detection, resulting in the errors illustrated in Figure 10b. Lastly, both methods employ limited geometric constraints from the N-wire phantom and did not effectively suppress non-feature points, contributing to the false detections shown in Figure 10e,f.

While the processing time of our method is slightly longer—by approximately 1.7 to 1.9 s per image group—this increase is acceptable in clinical practice. More importantly, our method offers significantly improved detection accuracy and reliability. These results demonstrate that our approach provides substantial practical advantages over the existing methods, and the incorporation of geometric constraints represents a key innovation in feature point detection for ultrasound calibration.

## 4. Discussion

To ensure both accuracy and robustness in feature point extraction, this study proposes a fully automated strategy that leverages the structural priors of the N-wire phantom by integrating imaging-based and geometric-based constraints. The proposed approach comprises two main components. From the imaging perspective, due to the ultrasound beam’s point spread function, the embedded nylon wires in the N-wire phantom appear as localized high-intensity regions in the ultrasound images. These bright spots exhibit distinctive intensity and morphological characteristics, which serve as shape priors for feature detection. To enhance the saliency of these features, a top-hat morphological transformation is applied to suppress large-scale background interference and emphasize the localized wire signals. A multi-scale Hessian matrix analysis is subsequently employed to enhance the feature response across varying imaging depths, thereby facilitating robust detection of candidate feature points. On the other hand, from the geometric perspective, the N-wire phantom exhibits a highly regular, mesh-like structure in which feature points demonstrate strong intra-layer collinearity, inter-layer parallelism, and consistent spatial spacing. These geometric priors are used to constrain the selection of candidate feature point combinations, allowing the algorithm to filter out spurious detections that deviate from the expected structural configuration. By integrating these imaging and geometric constraints, the proposed method effectively suppresses false positives and significantly improves the stability and accuracy of feature localization.

The experimental results demonstrated that the proposed extraction method achieved high recognition rates ranging from 97.333% to 99.000% across all tested imaging depths, with a consistent false detection rate of 0%, even when operated by different users. This confirms the algorithm’s robustness with respect to operator variability. Furthermore, the ablation studies underscored the complementary roles of the top-hat transformation, multi-scale Hessian enhancement, and structural constraint modules. Removal of any individual component resulted in a noticeable drop in detection performance, validating the synergistic effectiveness of the full pipeline.

In terms of calibration accuracy, this study introduces a RANSAC-based outlier rejection mechanism integrated with the L-M optimization algorithm. RANSAC iteratively selects random minimal subsets to estimate temporary transformation models, computes residuals for the remaining data, and identifies the set with the highest inlier ratio. Once a sufficient inlier set is identified—without requiring exhaustive enumeration—the L-M algorithm refines the calibration parameters using this filtered data. This hybrid approach balances computational efficiency with robustness. The quantitative evaluations revealed that RANSAC reduced the mean Euclidean distance error by 0.20–0.30 mm and the maximum error by 0.25–0.40 mm across all imaging depths, confirming its substantial contribution to the overall calibration precision.

Despite the effectiveness of the proposed automatic calibration framework, certain limitations remain. First, due to significant variations in image quality across different ultrasound devices and the inherent beam thickness, the algorithm may fail to accurately localize feature points in data acquired from unfamiliar systems, thereby affecting its generalizability. To mitigate this, future work should consider device-specific calibration profiling to improve system compatibility. Second, the point spread function (PSF) of the ultrasound beam can blur the intensity peaks of feature points, making precise center localization challenging. Incorporating PSF modeling and image deconvolution techniques could enhance the spatial accuracy. Third, the current calibration process requires manual frame-by-frame image acquisition, which is labor-intensive and demands a high level of operator expertise. This limits the method’s efficiency and usability in clinical settings. As a future improvement, a keyframe detection mechanism based on continuous ultrasound video streams could be introduced. This would enable automatic selection of informative frames for calibration, significantly enhancing the data acquisition efficiency and reducing variability caused by different users.

## 5. Conclusions

This study addresses key challenges in ultrasound probe calibration by introducing a fully automated calibration method grounded in the structural priors of the N-wire phantom. To improve feature point extraction in the presence of speckle noise and blurred signals, the proposed approach incorporates multi-level constraints based on both shape and geometric structure priors, substantially enhancing the accuracy and robustness of feature localization. For coordinate transformation estimation, a RANSAC-based outlier rejection strategy is employed to suppress erroneous correspondences and enable high-precision estimation of the rigid transformation matrix. Experimental validation across multiple independent trials confirmed that the method can achieve sub-millimeter accuracy at various imaging depths. Furthermore, the repeatability assessments demonstrated strong consistency and reliability across executions. In summary, the proposed method offers a robust, accurate, and fully automated solution for ultrasound probe calibration, eliminating the need for manual intervention and providing a reliable foundation for the development of ultrasound-guided surgical navigation systems.

## Figures and Tables

**Figure 1 sensors-25-05104-f001:**
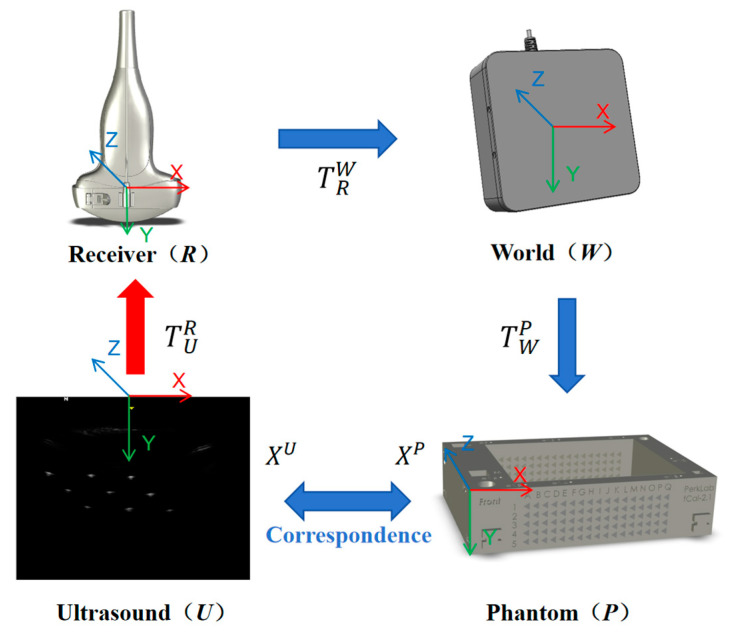
Coordinate transformation relationship diagram. The geometry of the N-wire phantom helps to bridge the feature points in the ultrasound image with their physical positions in the phantom space, through which a closed-form equation can be established.

**Figure 2 sensors-25-05104-f002:**
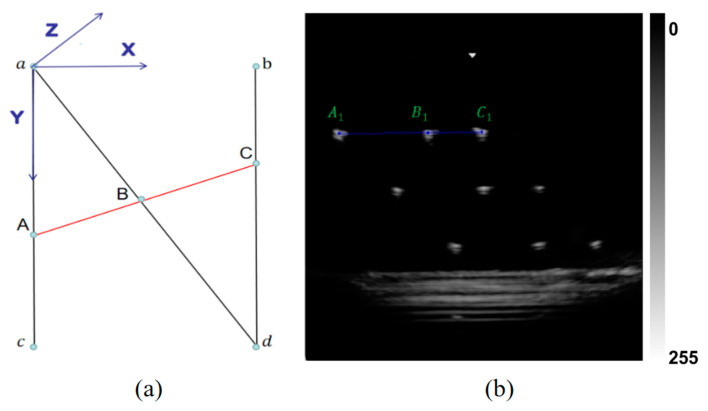
N-wire phantom geometric structure and ultrasound imaging diagram. (**a**) Schematic of the N-wire phantom geometric structure; (**b**) Corresponding ultrasound imaging diagram.

**Figure 3 sensors-25-05104-f003:**
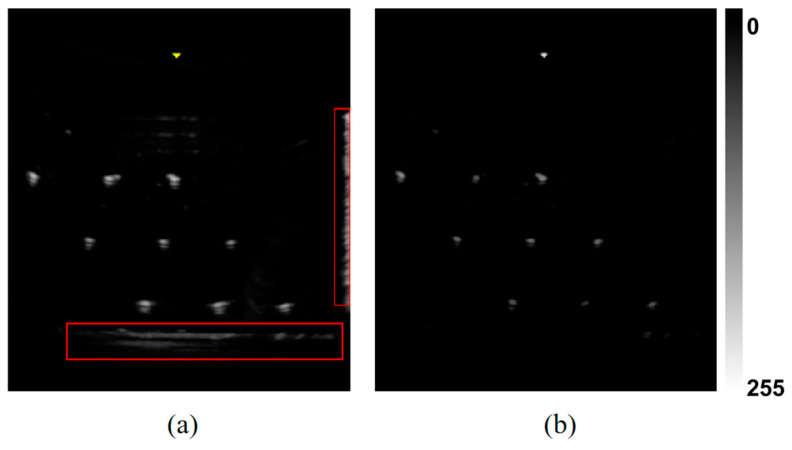
Top-hat transformation results: (**a**) original ultrasound image with highlighted interference regions (red rectangles indicate elongated echo artifacts caused by structural reflections); (**b**) processed image after top-hat filtering.

**Figure 4 sensors-25-05104-f004:**
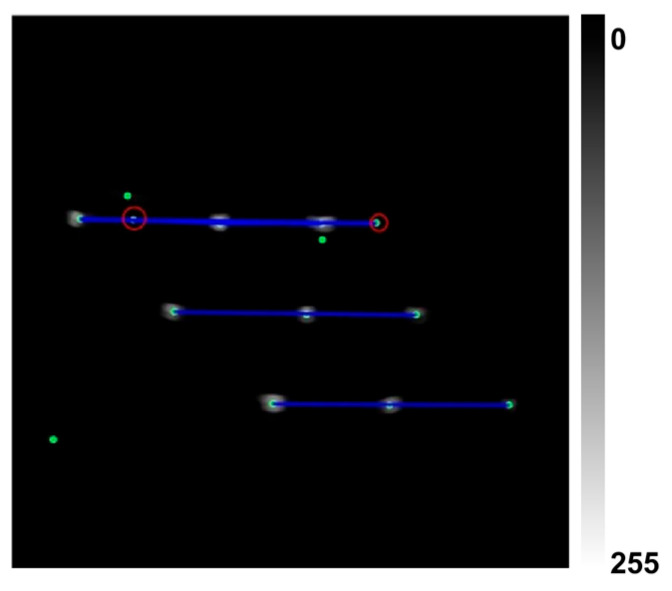
Geometric constraint-based feature point filtering: green dots indicate candidates, blue lines indicate valid lines, and red circles indicate rejected outliers.

**Figure 5 sensors-25-05104-f005:**
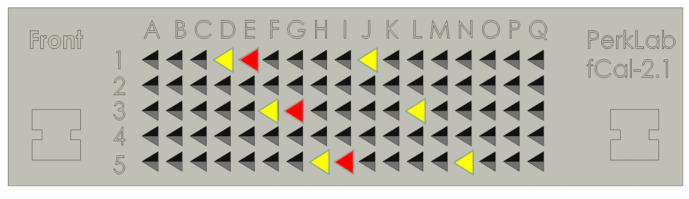
Arrangement of parallel and oblique wires in the N-wire phantom. Yellow triangles represent parallel nylon wires, and red triangles denote oblique lines.

**Figure 6 sensors-25-05104-f006:**
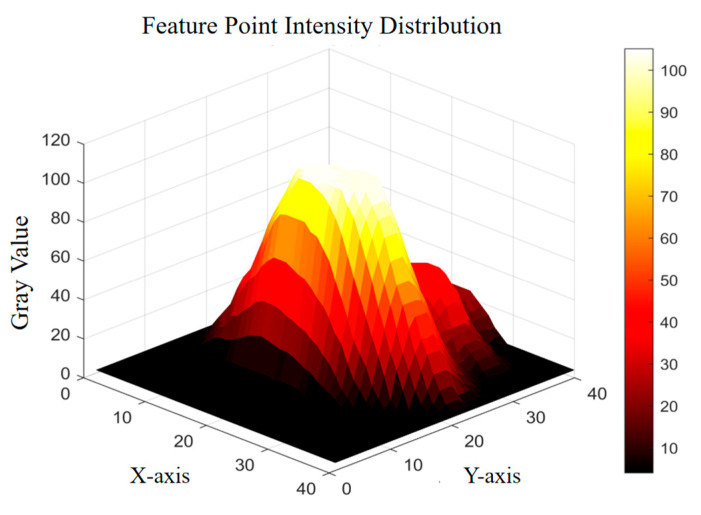
Intensity distribution map of a feature point.

**Figure 7 sensors-25-05104-f007:**
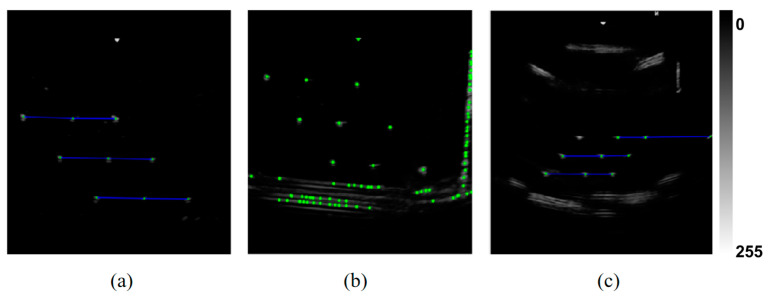
Comparison of feature point recognition under different processing configurations. (**a**) Recognition result with full algorithm pipeline: feature points correctly identified under structural constraints; (**b**) result without top-hat preprocessing: strong echoes from large structures lead to false positives; (**c**) result without geometric distance constraint: collinear points retained, but invalid due to missing inter-layer spacing verification.

**Figure 8 sensors-25-05104-f008:**
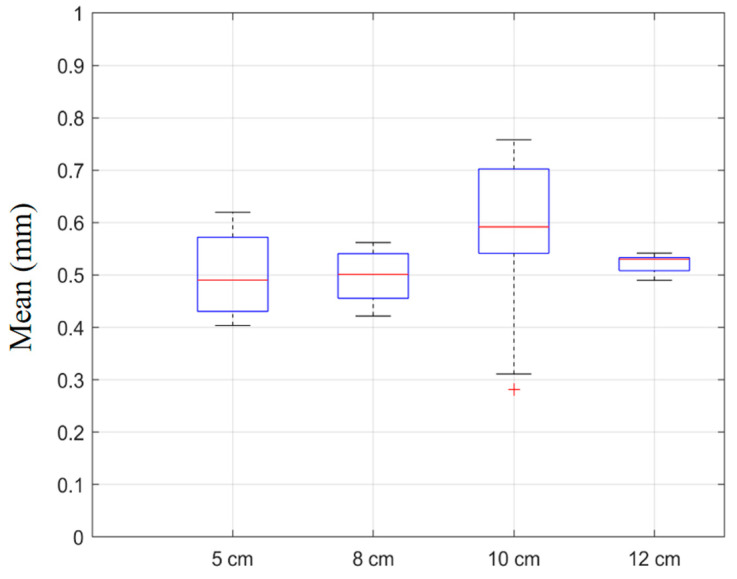
Boxplot of calibration error for the proposed automatic method across different imaging depths.

**Figure 9 sensors-25-05104-f009:**
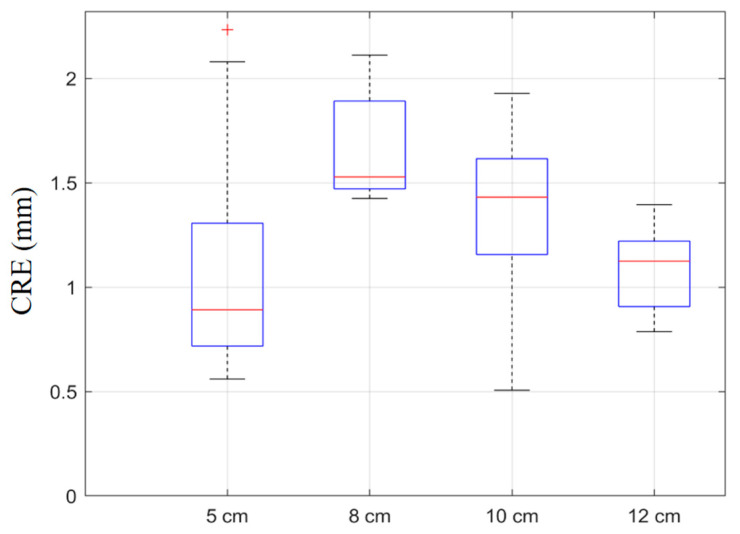
Boxplot of calibration reproducibility error (*CRE*) at different imaging depths.

**Figure 10 sensors-25-05104-f010:**
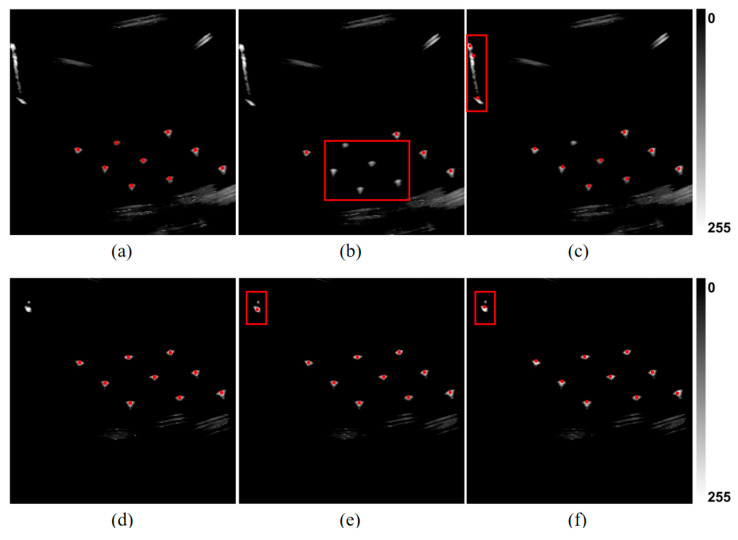
Comparison of detection results: (**a**–**c**) and (**d**–**f**) show detection results on the same ultrasound image using different methods. From left to right: (**a**,**d**) our method; (**b**,**e**) AGC-FD; (**c**,**f**) TACD. Red boxes indicate examples of missed detections or false detections.

**Table 1 sensors-25-05104-t001:** Recognition performance and processing time of the proposed method across different ultrasound systems and imaging depths.

Ultrasound System	Probe Type	Depth (cm)	Recognition Rate (%)	Processing Time (s)
NeuBook3	Convex array	5	97.333	11.06945
		8	98.333	12.37848
		12	98.215	12.19984
	Linear array	5	98.333	9.84532
		8	97.667	10.26384
		12	99.000	10.78921
Philips EPIQ 5	Convex array	5	98.667	9.4497
		8	97.000	12.2214
		12	98.667	14.3551
	Linear array	5	93.333	11.2395
		8	94.215	15.7213
		12	98.215	10.2704

**Table 2 sensors-25-05104-t002:** Comparison of recognition rates and false detection rates under different combinations of recognition modules.

Top-Hat Transformation	Multi-Scale Hessian	Geometric Constraint	Depth (cm)	Recognition Rate (%)	False Detection Rate (%)
√	√	√	5	97.333	/
			12	98.215	/
√		√	5	94.000	1.050
			12	80.333	2.083
√	√		5	87.333	10.344
			12	86.000	10.417

**Table 3 sensors-25-05104-t003:** Calibration error statistics under different module combinations and scanning depths.

L-M	RANSAC	Depth (cm)	Mean (mm)	Max (mm)
√	√	5	0.4979	0.6196
		8	0.4986	0.5618
		10	0.5709	0.7580
		12	0.5241	0.5419
√		5	0.7297	0.8154
		8	0.7458	0.7789
		10	0.8679	0.9958
		12	0.8879	1.0214

**Table 4 sensors-25-05104-t004:** Calibration reproducibility metric statistics at different scanning depths.

Depth (cm)	Mean (mm)	Max (mm)	Min (mm)
5	1.121	2.2330	0.5601
8	1.672	2.1047	1.4250
10	1.385	1.7384	0.5061
12	1.092	1.3952	0.7866

**Table 5 sensors-25-05104-t005:** Quantitative comparison of feature point detection performance of different methods.

Data Group	Method	Recognition Rate (%)	False Detection Rate (%)	Processing Time (s)
1	Ours	100	0	8.87
	AGC-FD	90.69	8.23	7.31
	TACD	94.63	5.37	7.06
2	Ours	96.43	0	9.38
	AGC-FD	89.84	9.79	7.49
	TACD	90.29	8.33	7.25
3	Ours	98.04	0	6.35
	AGC-FD	94.81	5.81	4.52
	TACD	95.30	4.69	4.59

## Data Availability

The datasets generated during this study are derived from experiments conducted by the authors. The data are not publicly available but can be obtained from the corresponding author upon reasonable request.

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
