# Peer review of "A Novel Shape-Prior-Guided Automatic Calibration Method for Free-Hand Three-Dimensional Ultrasonography"

_sensors, 2025, doi:10.3390/s25165104_

Round 1

Reviewer 1 Report

Comments and Suggestions for Authors

    The authors present a novel automatic calibration method for free-hand 3D ultrasonography, which introduces a moving segmentation approach to improve calibration efficiency. While the method demonstrates promising results, its comprehensive performance requires further validation. Below are specific suggestions for improvement: 

  1. Figure 1 Consistency: The coordinate transformation directions in Figure 1 are inconsistent. Please revise the figure to ensure uniformity.
  2. Figure 1 Clarity: The XYZ axes in Figure 1 should be explicitly labeled. Consider using a three-color coordinate axis system for better visualization.
  3. System Specifications: To ensure reproducibility, please provide detailed specifications of the experimental setup, including:
  •    Ultrasound device and probe models. 
  •    Electromagnetic tracking device model and resolution. 
  •    Physical dimensions of the N-wire phantom. 
  1. Calibration Reproducibility Error (CRE): The reported CRE performance is suboptimal. Please address potential causes and suggest improvements.
  2. Figure and Table Formatting:
  • Figures 6, 8, and 9 are excessively large; consider resizing them. 
  • Table row spacing should be compressed for better readability. 
  1. Language and Algorithm Description:
  • The manuscript’s language could be more concise. 
  • Descriptions of the Levenberg–Marquardt (L-M) and RANSAC algorithms can be streamlined by referencing prior literature. 
  1. References and Benchmarking:
  • Include more recent references (last 5 years) to contextualize the advancements in ultrasound calibration. 
  • Explicitly compare the proposed method’s accuracy with existing algorithms to highlight its contributions. 

    Overall, this work has significant potential, but addressing these points would strengthen its rigor and clarity. 

Author Response

Comments 1: Figure 1 Consistency: The coordinate transformation directions in Figure 1 are inconsistent. Please revise the figure to ensure uniformity.
Response 1: Thank you for your careful review and for highlighting the potential confusion regarding the transformation directions in Figure 1. We respectfully clarify that the transformation notation  used in our manuscript denotes a mapping from coordinate system  to , and this convention has been explicitly stated in the text (see Section 2.1, paragraph 2).
To enhance clarity and address your concern, we have revised Figure 1 to ensure that all transformation arrows follow a consistent direction, visually aligning with the defined transformation logic. In addition, to better represent the closed-loop transformation structure underlying our calibration framework, we have added a bidirectional arrow between the Ultrasound (U) and Phantom (P) coordinate systems, indicating the existence of corresponding feature points in both frames. This addition emphasizes the mutual spatial correspondence that supports the optimization formulation described in Equation (1).
Moreover, the revised structure of Figure 1 is now consistent with the widely recognized convention used in Chen et al. (A Real-Time Freehand Ultrasound Calibration System with Automatic Accuracy Feedback and Control, Ultrasound in medicine & biology, 2009), particularly the coordinate relationship illustrated in their Figure 9. We believe this consistency with established literature further strengthens the clarity and interpretability of our figure.
The updated figure has also been refined for better visual clarity and consistency. We sincerely thank you for your constructive feedback, which helped improve the figure’s quality and readability.

This is the updated Figure 1:

Fig 1. Coordinate transformation relationship diagram. The geometry of  N-wire phantom helps to bridge the feature points in the ultrasound image with their physical positions in the phantom space, through which a closed-form equation could be established.

Comments 2: Figure 1 Clarity: The XYZ axes in Figure 1 should be explicitly labeled. Consider using a three-color coordinate axis system for better visualization.
Response 2: Thank you for your valuable suggestion to improve the clarity of Figure 1. In response, we have explicitly labeled the X, Y, and Z axes for each coordinate system in the diagram. Furthermore, we have adopted a standard three-color scheme to enhance visual distinction: red for the X-axis, green for the Y-axis, and blue for the Z-axis.
These changes help improve the interpretability of the coordinate transformations and provide better visual guidance for readers. The updated Figure 1 is included in our response to Comment 1 and also reflected in the revised manuscript

Comments 3: System Specifications: To ensure reproducibility, please provide detailed specifications of the experimental setup, including:
   Ultrasound device and probe models. 
   Electromagnetic tracking device model and resolution. 
   Physical dimensions of the N-wire phantom. 
Response 3: Thank you for your thoughtful suggestion regarding experimental reproducibility. In response, we have added a detailed description of the system specifications at the beginning of Section 2.4 (Experiment Design) in the revised manuscript. The information includes:

  • Two ultrasound imaging systems were used in this study: the Philips EPIQ 5 (Philips Healthcare, Amsterdam, The Netherlands) and the NeuBook 3 (Neusoft Medical Systems Co., Ltd., Shenyang, China).
  • The models of the probes: convex array (C5-2) and linear array (L12-5).
  • The video capture device: HD60 S+ (HDMI/USB 3.0), with a capture frame rate of 60 fps.
  • The electromagnetic tracking system: NDI Aurora (Northern Digital Inc., Canada), operating at 40 Hz, with a spatial tracking accuracy better than 1 mm. A micro sensor was rigidly mounted on the ultrasound probe.
  • The N-wire phantom: fabricated using a 3D-printed open-source design from the Plus Toolkit project, with a vertical layer spacing of 10 mm, horizontal spacing between parallel wires of 25 mm, and embedded nylon wires with a diameter of 0.3 mm.
  • The computational platform: a Windows 11 (64-bit) system equipped with an Intel Core i7-12700H CPU, an NVIDIA RTX 3060 GPU, and 16 GB of RAM. The software environment includes C++, OpenCV, Eigen, MATLAB, and the NDI SDK.

These additions ensure transparency and facilitate experimental replication by other researchers.

Comments 4: Calibration Reproducibility Error (CRE): The reported CRE performance is suboptimal. Please address potential causes and suggest improvements.
Response 4: Thank you for your constructive comment regarding the Calibration Reproducibility Error (CRE). We acknowledge that, while the overall performance of the proposed method remains acceptable for practical use, the CRE values could be further improved. We believe the following two factors are the main contributors to the observed deviations:
1.Operator-induced variability in data acquisition:
During manual ultrasound scanning, different operators may apply varying levels of pressure, scanning angles, or probe motion patterns. These inconsistencies can lead to variations in image quality and feature visibility, thereby affecting the calibration accuracy and inter-trial consistency.
2.Stochastic nature of the RANSAC algorithm:
The RANSAC-based optimization framework involves random sampling and does not always guarantee the globally optimal inlier set. Different random seeds or sampling combinations across calibration trials may lead to slightly different inlier subsets, resulting in variations in the estimated transformation matrices and, consequently, in CRE values.
To address these limitations and further improve the reproducibility of the calibration process, we propose the following future directions:
1.Automated frame selection from ultrasound video streams:
Instead of relying on manually acquired individual frames, a real-time keyframe selection algorithm can be integrated to automatically extract frames with optimal geometric diversity and image quality. This would reduce operator dependency and enhance calibration consistency.
2.Improved outlier rejection strategy:
We plan to explore more robust sampling and consensus mechanisms, such as guided sampling strategies or deterministic inlier maximization algorithms, to reduce the variance introduced by random sampling in the RANSAC process.
Thank you again for your insightful feedback.

Comments 5: Figure and Table Formatting:
Figures 6, 8, and 9 are excessively large; consider resizing them. 
Table row spacing should be compressed for better readability. 
Response 5: Thank you for your valuable feedback regarding the formatting of figures and tables. In response to your suggestions: Figures 6, 8, and 9 have been resized in the revised manuscript to reduce their visual footprint and improve layout compactness, while maintaining sufficient resolution and readability. Table row spacing has been compressed uniformly across all tables to improve overall presentation and enhance content density without compromising legibility.
These formatting adjustments contribute to a more concise and reader-friendly manuscript layout. We sincerely appreciate your attention to these important details.

Comments 6: Language and Algorithm Description:
The manuscript’s language could be more concise. 
Descriptions of the Levenberg–Marquardt (L-M) and RANSAC algorithms can be streamlined by referencing prior literature. 
Response 6: Thank you for your helpful suggestion. In response, we have revised the description of the optimization strategy in Section 2.3 (Robust Optimization via RANSAC-Assisted Levenberg–Marquardt) to improve conciseness while maintaining the logical flow and technical clarity.
Specifically, we have removed redundant explanations regarding the basic principles of the Levenberg–Marquardt (L-M) and RANSAC algorithms and replaced them with more compact descriptions. At the same time, we retained the essential steps of the algorithm to ensure reader comprehension. To avoid duplication, the mathematical notations and variable definitions used in Equation (9) now reference Equation (1), where they are first introduced in detail.
These changes enhance readability and align the manuscript with standard academic conventions by relying on well-established references for algorithmic details.

Comments 7: References and Benchmarking:
Include more recent references (last 5 years) to contextualize the advancements in ultrasound calibration. 
Explicitly compare the proposed method’s accuracy with existing algorithms to highlight its contributions. 
Response 7: Thank you for your valuable suggestion regarding the inclusion of more recent references to contextualize advancements in ultrasound calibration.
In response to the first suggestion, we have revised the Introduction section (see the last two paragraphs) to incorporate several representative studies from the past five years that reflect ongoing efforts to improve the automation, accuracy, and robustness of ultrasound calibration. These include the work of Wu et al. [12], who proposed a fully automatic method using a multilayer N-wire phantom; Shen et al. [19], who developed a flexible wire phantom-based approach relying on manual feature identification; and recent robot-assisted calibration methods that utilize geometric phantoms such as sphere phantoms [20], Z-fiducial phantoms [21], and M-fiducial phantoms [22].
We have also analyzed the limitations of these approaches, including their susceptibility to false detections due to simple geometric fitting, the need for manual intervention in stylus-based methods [20][23], and the general lack of robust outlier rejection mechanisms [13][22][23]. These limitations emphasize the need for more reliable, fully automated calibration strategies.
By integrating this analysis, we aim to better position our proposed method within the current research landscape and highlight its contributions in addressing the aforementioned challenges. The relevant references have been added to the manuscript accordingly.

In response to the second suggestion, we have added a comprehensive comparison of feature point detection accuracy in Section 3.4. We implemented two representative N-wire phantom-based methods for comparison:
1.AGC-FD (Automatic Gray-weighted Centroid Feature Detection), a fully automated approach, and
2.TACD (Threshold-Assisted Centroid Detection), a semi-automated method.
All three methods, including our proposed method, were evaluated on three independent ultrasound datasets. As summarized in Table 5, our method achieved the highest average recognition rate (98.16%) and zero false detections, significantly outperforming AGC-FD and TACD, which achieved average recognition rates of 91.87% and 93.41%, respectively, with false detection rates of 7.94% and 6.08%. These results provide strong quantitative evidence of the superior accuracy and reliability of our method.
Additionally, although our method incurred slightly longer processing times (by approximately 1.7 to 1.9 seconds per image group), this is acceptable in clinical settings given the substantial accuracy improvements.
Furthermore, Figure 10 provides visual comparisons illustrating common failure modes of AGC-FD and TACD, including missed detections and false positives caused by their limited handling of ultrasound artifacts and lack of geometric feature enforcement. 
This comparison explicitly demonstrates the novelty and effectiveness of our approach, particularly the integration of shape priors and geometric constraints in feature point detection, which is not addressed by existing methods.
Finally, I have pasted Table 5 and Figure 10 below for your review. We appreciate the reviewer’s suggestion, which helped us better emphasize the methodological contributions and performance advantages of our work.

Reviewer 2 Report

Comments and Suggestions for Authors

This paper addresses critical challenges in ultrasound probe calibration, focusing on improving accuracy, robustness, and efficiency through automation.

Ensure that all figures referenced in the text, e.g., Fig 6 is not properly included in the text.

Please nclude a quantitative comparison of your method's performance metrics (accuracy, false detection rate, processing time) against at least one or two representative state-of-the-art automated or semi-automated ultrasound calibration methods. This will provide strong evidence for the novelty and effectiveness claimed.

Please Provide more concrete strategies or preliminary results regarding the adaptation of the method to different ultrasound devices, beyond just stating it as future work.

Author Response

Comments 1: Ensure that all figures referenced in the text, e.g., Fig 6 is not properly included in the text.
Response 1: We sincerely thank the reviewer for pointing out the inconsistency regarding the figure references in the text. After careful review, we found that this was indeed a typographical error on our part.
Specifically, in Section 2.2 (3) Subpixel Center Localization of Feature Points, the figure referenced as “Fig. 7” should have been “Fig. 6”. This section describes the method for refining the feature point positions using subpixel intensity analysis, where a search window is used to locate the pixel with the highest local average intensity. The visual illustration corresponding to this procedure is provided in Fig. 6, titled “Intensity distribution map of feature point”. This figure clearly depicts the intensity profile and the pixel neighborhood used for final center localization, which aligns directly with the described method.
On the other hand, Fig. 7 is actually referenced in Section 3.1 as part of an ablation comparison, not the subpixel localization step. Therefore, the original reference to Fig. 7 in the subpixel localization paragraph was incorrect.
We have corrected this mistake in the revised manuscript to accurately reference Fig. 6 in this section. We truly appreciate the reviewer’s careful reading, which helped us improve the clarity and accuracy of the manuscript.

Comments 2: Please include a quantitative comparison of your method's performance metrics (accuracy, false detection rate, processing time) against at least one or two representative state-of-the-art automated or semi-automated ultrasound calibration methods. This will provide strong evidence for the novelty and effectiveness claimed.
Response 2: We sincerely thank the reviewer for the insightful suggestion to provide a quantitative comparison with representative state-of-the-art ultrasound calibration methods, which is crucial for demonstrating the novelty and effectiveness of our approach.
In response, we have added a new section (Section 3.4 Comparison with Representative Methods) to the manuscript. In this section, we implemented and evaluated two representative N-wire phantom-based methods:
1.AGC-FD (Automatic Gray-weighted Centroid Feature Detection), a fully automated method proposed by Rong et al., which utilizes adaptive thresholding and gray-level weighted centroid extraction. While automated, this method lacks geometric constraints and is sensitive to noise, resulting in higher false detection rates.
2.TACD (Threshold-Assisted Centroid Detection), a semi-automated method derived from the work of Lei et al., which relies on threshold-based segmentation and centroid detection via minimum enclosing circle fitting. Threshold selection in TACD often requires manual adjustment and the method similarly does not incorporate geometric priors.
All methods were evaluated using three independent datasets of ultrasound images acquired at different imaging depths (5 cm, 8 cm, and 12 cm), with performance assessed in terms of recognition rate, false detection rate, and processing time per image group.
As summarized in Table 5, our proposed method achieved the highest average recognition rate (98.16%) with 0% false detection rate across all datasets. In comparison, AGC-FD and TACD exhibited significantly lower recognition rates (91.87% and 93.41%) and higher false detection rates (7.94% and 6.08%, respectively). Although our method incurred a modest increase in processing time (1.7 to 1.9 seconds per image group), this is acceptable in clinical practice given the substantial improvement in detection accuracy and reliability.
Furthermore, Figure 10 provides visual comparisons illustrating common failure modes of AGC-FD and TACD, including missed detections and false positives caused by their limited handling of ultrasound artifacts and lack of geometric feature enforcement. In contrast, our method leverages the geometric priors of the N-wire phantom for robust and accurate feature point detection, which represents a key innovation.
Finally, I have pasted Table 5 and Figure 10 below for your review. We appreciate the reviewer’s valuable guidance, which prompted us to strengthen our experimental validation and more clearly demonstrate the practical advantages and novelty of our approach.

Comments 3: Please Provide more concrete strategies or preliminary results regarding the adaptation of the method to different ultrasound devices, beyond just stating it as future work.
Response 3: We thank the reviewer for the insightful suggestion to provide more concrete strategies or preliminary results regarding the adaptation of our method to different ultrasound devices.
In response, we have conducted additional experiments using a Philips EPIQ 5 ultrasound system, complementing the original experiments performed with the NeuBook 3 system. These experiments evaluated feature point detection performance at three representative imaging depths (5 cm, 8 cm, and 12 cm). Based on our prior findings that depth variation has limited impact on detection accuracy, we streamlined the experimental design by omitting the 10 cm depth in this supplementary study.
The results, summarized in the new Table 1, demonstrate that the proposed algorithm achieves consistently high recognition rates and stable processing times across both devices. This confirms the method’s generalizability and adaptability to different ultrasound systems.
The manuscript has been revised accordingly, with updated descriptions in Section 3.1 and a replacement of the original Table 1. We appreciate the reviewer’s suggestion, which has strengthened the practical relevance and rigor of our study. For clarity, the updated Table 1 is provided below.

Reviewer 3 Report

Comments and Suggestions for Authors

  1. The titles of Sections 3.1 and 3.2 are the same. Please revise the title of 3.2 to reflect its actual content.
  2. All ultrasound images should include a grayscale (dynamic range) bar for clarity and reproducibility.
  3. The manuscript does not provide details about the imaging system, ultrasound probe, EM tracking system, or phantom used in the experiments. This information is necessary for reproducibility and evaluating applicability across devices. Please include it.
  4. Figure 6 is shown but not described in the text. Please add a brief explanation.
  5. Reference [20] should be placed at the end of the sentence where the Levenberg–Marquardt algorithm is mentioned.

Author Response

Comments 1: The titles of Sections 3.1 and 3.2 are the same. Please revise the title of 3.2 to reflect its actual content.
Response 1: We sincerely thank the reviewer for carefully pointing out the duplicated section title. After reviewing the manuscript, we confirmed that the title of Section 3.2 was mistakenly duplicated from Section 3.1, both appearing as “Feature Point Detection Accuracy”. This was a formatting oversight during manuscript editing, and we apologize for the confusion it may have caused. The correct title of Section 3.2 should be “Automatic Calibration Accuracy”, which more accurately reflects the actual content of this section. In this part of the manuscript, we evaluate the spatial accuracy of the proposed automatic calibration method by measuring the Euclidean distance between the transformed ultrasound feature points and their corresponding ground truth positions in the phantom coordinate system. We also compare the performance of our RANSAC-assisted optimization against a conventional Levenberg–Marquardt-only approach across varying imaging depths.
The corrected title properly represents this focus on the final calibration performance rather than feature point detection, and ensures consistency with the structure of Section 3.3 (“Automatic Calibration Repeatability”).
We have revised the manuscript accordingly and sincerely appreciate the reviewer’s attention to detail, which helped improve the clarity and accuracy of our presentation.

Comments 2: All ultrasound images should include a grayscale (dynamic range) bar for clarity and reproducibility.
Response 2: We sincerely thank the reviewer for the valuable suggestion regarding the inclusion of grayscale (dynamic range) bars in the ultrasound images to enhance clarity and reproducibility.
In response to this comment, we have carefully revised Figures 2, 3, 4, and 7 by adding grayscale bars on the right side of each image. These bars reflect the actual intensity distribution of the corresponding original or processed images and serve as visual references for interpreting brightness and contrast levels. We believe this modification improves the interpretability and reproducibility of the presented results.
We appreciate the reviewer’s attention to detail, which helped us improve the quality and clarity of the figures.

Comments 3: The manuscript does not provide details about the imaging system, ultrasound probe, EM tracking system, or phantom used in the experiments. This information is necessary for reproducibility and evaluating applicability across devices. Please include it.
Response 3: Thank you for your thoughtful suggestion regarding experimental reproducibility. In response, we have added a detailed description of the system specifications at the beginning of Section 2.4 (Experiment Design) in the revised manuscript. The information includes:

  • Two ultrasound imaging systems were used in this study: the Philips EPIQ 5 (Philips Healthcare, Amsterdam, The Netherlands) and the NeuBook 3 (Neusoft Medical Systems Co., Ltd., Shenyang, China).
  • The models of the probes: convex array (C5-2) and linear array (L12-5).
  • The video capture device: HD60 S+ (HDMI/USB 3.0), with a capture frame rate of 60 fps.
  • The electromagnetic tracking system: NDI Aurora (Northern Digital Inc., Canada), operating at 40 Hz, with a spatial tracking accuracy better than 1 mm. A micro sensor was rigidly mounted on the ultrasound probe.
  • The N-wire phantom: fabricated using a 3D-printed open-source design from the Plus Toolkit project, with a vertical layer spacing of 10 mm, horizontal spacing between parallel wires of 25 mm, and embedded nylon wires with a diameter of 0.3 mm.
  • The computational platform: a Windows 11 (64-bit) system equipped with an Intel Core i7-12700H CPU, an NVIDIA RTX 3060 GPU, and 16 GB of RAM. The software environment includes C++, OpenCV, Eigen, MATLAB, and the NDI SDK.

These additions ensure transparency and facilitate experimental replication by other researchers.

Comments 4: Figure 6 is shown but not described in the text. Please add a brief explanation.
Response 4: We sincerely thank the reviewer for pointing out that Figure 6 was shown but not described in the text. After careful review, we found that this issue was due to a typographical error in Section 2.2(3) Subpixel Center Localization of Feature Points, where the figure reference was incorrectly written as “Fig. 7” instead of “Fig. 6”. Specifically, this section describes the method for refining the feature point positions using subpixel intensity analysis, where a search window is used to locate the pixel with the highest local average intensity. The visual illustration corresponding to this procedure is provided in Fig. 6, titled “Intensity distribution map of feature point”. This figure clearly depicts the intensity profile and the pixel neighborhood used for final center localization, which aligns directly with the described method.
On the other hand, Fig. 7 is actually referenced in Section 3.1 as part of an ablation comparison, not the subpixel localization step. Therefore, the original reference to Fig. 7 in the subpixel localization paragraph was incorrect.
We have corrected this mistake in the revised manuscript to accurately reference Fig. 6 in this section. We truly appreciate the reviewer’s careful reading, which helped us improve the clarity and accuracy of the manuscript.

Comments 5: Reference [20] should be placed at the end of the sentence where the Levenberg–Marquardt algorithm is mentioned.
Response 5: We thank the reviewer for the helpful suggestion regarding the citation placement for the Levenberg–Marquardt algorithm.
As recommended, we have revised the manuscript by moving the relevant reference to the end of the sentence where the Levenberg–Marquardt (L-M) algorithm is first introduced, in Section 2.1 (second paragraph). However, due to the addition of several new references during the revision process, the citation number for the Levenberg–Marquardt algorithm has been updated from [20] to [25]. We have now correctly cited it as Reference [25] at the end of the corresponding sentence.
Additionally, we noticed that both “L-M” and “LM” were used inconsistently throughout the manuscript. To maintain consistency and improve clarity, we have standardized all mentions of the Levenberg–Marquardt algorithm to “L-M” throughout the text. All corresponding modifications have been marked in red font in the revised manuscript to facilitate review.
We appreciate the reviewer’s attention to detail, which has helped us improve the precision and consistency of the manuscript.

Round 2

Reviewer 1 Report

Comments and Suggestions for Authors

All comments are responded. It is suggested to accept this work. 

Reviewer 2 Report

Comments and Suggestions for Authors

Thank you to the authors for addressing the comments. The revised version shows significant improvement.